# Reproducibility Report for "On Warm-Starting Neural Network Training"

## Reproducibility Summary

**Scope of Reproducibility**

We reproduce the results of the paper "On Warm-Starting Neural Network Training." In many real-world applications, the training data is not readily available and is accumulated over time. As training models from scratch is a time-consuming task, it is preferred to use warm-starting, i.e., using the already existing models as the starting point to obtain faster convergence. This paper investigates the effect of warm-starting on the final model's performance. It identifies a noticeable gap between warm-started and randomly-initialized models, hereafter referenced as **the warm-starting gap**. Furthermore, they provide a solution to mitigate this side-effect. In addition to reproducing the original paper's results, we propose an alternative solution and assess its effectiveness.

**Methodology**

We reproduced almost every figure and table in the main text and some of those in the appendix. We used our implementation to produce these results. In case of a mismatch of the results, we also investigated the cause and proposed possible explanations. We mainly used GPUs to train our models using infrastructure offered by public clouds and those that were available to us privately.

**Results**

Most of our results closely match the reported results in the original paper. Therefore, we confirm that the warm-starting gap exists in certain settings and that the Shrink-Perturb method successfully reduces or eliminates this gap. However, in some cases, we were not able to completely reproduce their results. By investigating the root of such mismatches, we provide another solution to avoid this gap. In particular, we show that data augmentation also helps to reduce the warm-starting gap.

**What was easy**

The experiments described in the paper were based on regular training of neural networks on a portion of widely-used datasets, possibly from a pre-trained model. Therefore implementing each experiment was relatively easy to do. Furthermore, since many of the parameters were reported in the original paper, we did not need much tuning in most experiments. Finally, it is straightforward to implement and use the proposed solution.

**What was difficult**

Though implementing each experiment is relatively simple, the numerosity of experiments proved to be slightly challenging. In particular, each of the online experiments in the original setting requires training a deep network to convergence more than 30 times. In these cases, we sometimes changed the settings, sacrificing granularity to reduce computation time. However, these changes did not affect the interpretability of the final results.

**Communication with original authors**

We briefly communicated with the authors to clarify the experiments' details, such as the convergence conditions.

# 1 Introduction

Training large models from scratch is usually time and energy-consuming, so it is desired to have a method to accelerate retraining neural networks with new data added to the training set. The well-known solution to this problem is warm-starting. Warm-Starting is the process of using the weights of a model, pre-trained on a subset of the data, as the starting point of training with the complete data.

The paper investigates the effect of warm-starting on the final model's accuracy and identifies a generalization gap on warm-started models. The paper also provides a method to mitigate this gap by shrinking the pre-trained weights and adding a random perturbation.

In this report, we repeat the original paper's experiments and compare them with the reported results. Also, we extend the original paper results by investigating the effect of data augmentation on this phenomenon. In particular, we establish that using data augmentation might be a second solution to mitigating the generalization gap.

We report and discuss our results in Section 2. In section 3, we detail our experimental settings and hyperparameters.

# 2 Results & Discussion

## 2.1 Warm-Starting Generalization Gap

Similar to the paper, we start by demonstrating the existence of a generalization gap when using warm-starting before training. We use the same set of experiments used by the authors. Unless otherwise stated, we follow the settings described in the paper for our experiments.

In particular, in the offline setting, we first train our model on half of the training data and then further train the pre-trained model on the whole dataset. We compare the resulting model with a model trained from randomly initialized weights. Figure 1 depicts the test accuracy of ResNet-18 [1] during training in this setting and matches Figure 1 of the paper.

We repeat this experiment with different datasets, models, and optimizers. In particular we perform experiments on CIFAR-10 [2], CIFAR-100 [2], and SVHN [3]. As our model, we experiment with ResNet-18, a three-layer perceptron, and logistic regression. The same models and datasets were used in the original paper. For the optimizers, we compare SGD [4] and Adam [5]. In this particular experiment, we compare SGD with and without momentum. In the rest of this work, unless explicitly stated, SGD is used without momentum. The final accuracies are reported in Table 1 similar to Table 1 of the original paper. Our results are similar to the paper's results for CIFAR-10 and CIFAR-100 datasets. In particular, we observe a generalization gap when using a warm-started model instead of training from scratch. However, we did not observe the same gap on the SVHN dataset. Furthermore, we were unable to obtain a reasonable accuracy with the MLP model using SGD without momentum with the reported setting on SVHN. We instead report the result of using 0.005 learning rate. We did not perform hyperparameter tuning for the other experiments.

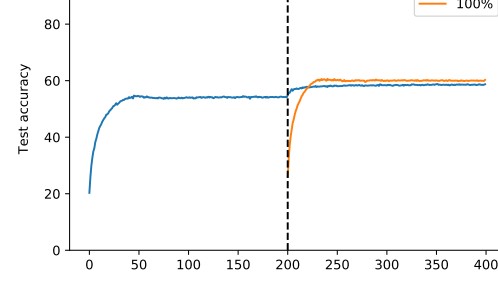

Figure 1: Test accuracy during training a ResNet-18 with SGD with and without warm-starting. The results for the randomly-initialized model has been shifted 200 epochs to overlap the part of training where the warm-started model is trained on the whole dataset.

In the online setting, we follow the original paper and train our model in several steps, increasing the amount of data available at each step. This setting is a more accurate simulation of the real-world problems where the training data grows over time.

We split the training data into batches of 1000 samples and start adding them, one by one, to the pool of available data. We follow two different scenarios. In one scenario, we reinitialize our model randomly after each batch is added and train it from scratch. In the other scenario, we continue training the model with the parameters learned in the previous step.

After each batch is added, we continue training our model until convergence before adding the next batch. We assume convergence when the model reaches 99% training accuracy. By communicating with the original paper's authors, we confirmed that this is the same condition used in the original paper.

As in the paper, we optimize the model using Adam optimizer with a learning rate of 0.001 on CIFAR-10. Given the discrepancy of our results on the SVHN datasets in the offline settings, we additionally perform the same experiment on this dataset. For the CIFAR-10 dataset, the generalization gap between random-initialization training and warm-start training is clearly observed (Figure 2a).

However, like the offline experiment, we did not observe the gap for the SVHN dataset (Figure 2b). Still, we were able to reproduce the gap by increasing the convergence accuracy threshold to $99.9\%$ (Figure 2c). Note that 99% train accuracy is more challenging to achieve on the CIFAR-10 dataset than on SVHN and therefore requires more training, possibly leading to more over-fitting. Increasing the convergence threshold compensates for this difference.

This result and the fact that the authors show that the proposed Shrink-Perturb method is similar to an aggressive regularization brings up the question of whether this gap might be a side-effect of over-fitting when training on partial data. There are various known techniques to prevent overfitting. The original paper investigates the effect of some of these techniques, namely regularization, and early-stopping. We reproduced these experiments and explained the results below.

**Early-Stopping**: To investigate the effect of early-stopping, following the original paper, we trained a ResNet-18 model on half of the CIFAR-10 data and checkpointed its parameters every 20 epochs. The result is plotted in Figure 3, which matches Figure 4 of the original paper and shows the warm-starting gap can be observed even after 20 epochs of training. To decrease computational costs, we used lower granularity than the original paper to perform this experiment, saving parameters every 20 epochs rather than 5 epochs. Also, we only perform each experiment once.

**Regularization**: Regularization is commonly used to improve generalization. The original paper explores the effect of various types of regularization. Due to time and resource limitations, we only look into weight decay, which is widely used and is a de-facto standard for training the state-of-the-art models. We repeat the offline setting experiment on CIFAR-10 with a weight decay of 0.1 on both the pre-training and main training. However, contrary to the original paper's results, we observe that the warm-starting gap decreases when applying weight decay. We also test with weight decay values of 0.01 and 0.001. We find out that higher values of weight decay result in lower warm-starting gap. The results are reported in Table 2, which corresponds to Appendix Table 13 of the original paper.

**Data Augmentation**: Data augmentation is widely used to obtain state-of-the-art performance and is known to help generalization [6], but it is not used in the other experiments of this paper. It is specifically important to check the effect of data augmentation since it is widely used in practice. Therefore we extend the original paper's experiments by investigating the impact of data augmentation. We report our results in Section 2.4.

## 2.2 Effect of Hyperparametes

Our results show that in some cases, a generalization gap exists when pre-training our model on a portion of the final dataset. However, when training in the online setting on SVHN, we could only observe this gap with a high enough convergence threshold. The paper investigates the effect of other training hyperparameters, namely learning rate and batch size.

To investigate the effect of learning rate and batch size, we train a ResNet-18 with different values for these hyperparameters. We choose the learning rate from $\{0.1, 0.01, 0.001\}$ and the batch size from $\{128, 64, 32, 16\}$. We iterate

| CIFAR-10 | SGD | ADAM | MSGD |
|---|---|---|---|
| Random Init | 60.3 (0.1) | 80.3 (0.1) | 65.5 (0.2) |
| Warm Start | 57.8 (0.4) | 79.0 (0.1) | 63.4 (0.2) |
| **SVHN** | | | |
| Random Init | 84.7 (0.1) | 92.4 (0.2) | 87.7 (0.1) |
| Warm Start | 86.3 (0.4) | 93.2 (0.2) | 87.2 (0.1) |
| **CIFAR-100** | | | |
| Random Init | 30.5 (0.3) | 48.8 (0.2) | 34.4 (0.1) |
| Warm Start | 27.6 (0.5) | 46.1 (0.2) | 30.6 (0.3) |

(a) Test accuracies for ResNet-18

| CIFAR-10 | SGD | ADAM | MSGD |
|---|---|---|---|
| Random Init | 38.2 (0.2) | 46.5 (0.2) | 45.9 (0.1) |
| Warm Start | 38.5 (0.3) | 46.0 (0.3) | 43.5 (0.4) |
| **SVHN** | | | |
| Random Init | 72.7(1.0)* | 72.2 (0.8) | 68.8 (1.0) |
| Warm Start | 67.3(1.7)* | 72.5 (0.5) | 70.0 (0.6) |
| **CIFAR-100** | | | |
| Random Init | 5.1 (0.2) | 19.5 (0.3) | 16.4 (0.1) |
| Warm Start | 5.1 (0.3) | 18.6 (0.1) | 16.6 (0.2) |

(b) Test accuracies for MLP

| CIFAR-10 | SGD | ADAM | MSGD |
|---|---|---|---|
| Random Init | 39.8 (0.1) | 35.6 (0.3) | 38.4 (0.3) |
| Warm Start | 39.7 (0.2) | 35.2 (0.3) | 38.5 (0.2) |
| **SVHN** | | | |
| Random Init | 19.8 (0.6) | 24.2 (0.9) | 22.4 (0.4) |
| Warm Start | 19.8 (0.4) | 24.4 (0.8) | 22.8 (0.4) |
| **CIFAR-100** | | | |
| Random Init | 16.6 (0.2) | 12.6 (0.2) | 17.2 (0.2) |
| Warm Start | 16.5 (0.1) | 12.2 (0.1) | 17.0 (0.1) |

(c) Test accuracies for Logistic Regression

Table 1: Test accuracies for various datasets and models and optimizer for warm-start training and training from random initialization. We use an MLP with Tanh activation with 3 hidden layers of 100 neurons. A different learning rate was used for cells marked with a star (*).

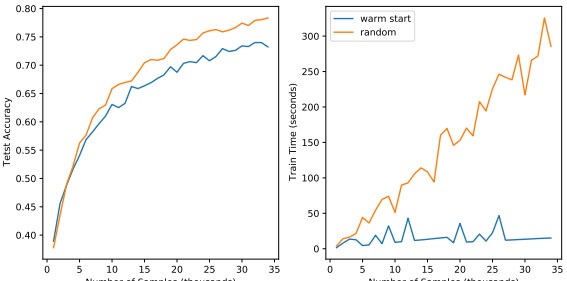 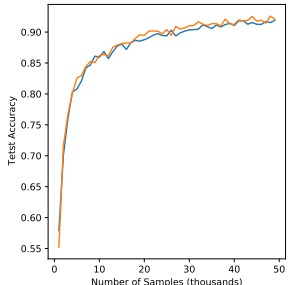 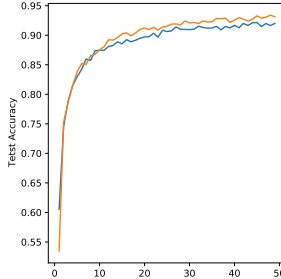

(a) For the CIFAR-10 dataset, the generalization gap between randomly initialized and warm started models is observed for the 99.0% convergence threshold. However, the training time required at each step increases almost linearly in the case of randomly initialized model, but is constant when using warm-starting.

(b) For the SVHN dataset, there is no generalization gap between randomly initialized and warm started models with the 99.0% convergence threshold.

(c) For SVHN, the generalization gap appears by increasing the convergence threshold to 99.9%.

Figure 2: Online learning experiment on CIFAR-10. the horizontal axis shows the number of samples available to train the model.

|  | 0.1 | 0.01 | 0.001 |
|---|---|---|---|
| Random Initialization | 81.44 | 62.73 | 60.42 |
| Warm Starting | 81.16 | 61.22 | 58.63 |

Table 2: Accuracy of training a ResNet-18 with and without warm-starting for different values of weight decay.

over all pairs for these values. For each pair, we train over the full CIFAR-10. We also train a different model over 50% of the CIFAR-10 dataset and use it to warm-start a model and train it on the whole dataset. We use a different learning rate and batch size, randomly chosen from the sets of values, in the second part of the training, i.e., for training the warm-started model. We repeat each experiment 9 times. Each model is trained to $99\%$ training accuracy. The test accuracy is plotted against training time in Figure 4, which corresponds with Figure 3 of the paper. Note that the training time for the warm-started model corresponds to the time of the second part of the training. In other words, the time of training on half of the dataset is not included. This is justified because the goal is to assess if warm-starting leads to comparable accuracy while saving training time when a new batch of data arrives. In our results, choosing the right hyperparameters can lead to achieving comparable or even better accuracy, when using warm-starting, faster than training a randomly initialized model. This does not match the results of the paper, where the warm-started models with comparable accuracy take the same amount of training time as the randomly initialized models. While we perform less experiments in the warm-started setting, we perform the same number of experiments with random initialization as described in the original paper's text. However, the number of points in Figure 3 of the original paper corresponding to randomly initialized models, is more than what has been described in the text. We note that performing more randomly initialized experiments might be the reason for the mismatch in our results with the original paper.

### 2.3 Shrink & Perturb Solution

In addition to establishing the warm-starting gap's existence and investigating its roots, the paper also provides a method to mitigate this issue. In this method, the training starts from a shrunk and perturbed version of the pre-trained weights, so we reference it as the Shrink-Perturb method. More specifically, for a given $\lambda$ and $\sigma$, the new weight is computed as

$$w_{new} = \lambda w_{pretrained} + \sigma w_{random} \tag{1}$$

where $w_{old}$ is the pre-trained weight and $w_{random}$ is the corresponding weight from a randomly initialized model. Whenever we apply the Shrink-Perturb transformation, we create a new randomly initialized model and use its weights as $w_{random}$.

We tested the effectiveness of this method in both offline and online settings. In the offline setting, we applied the Shrink-Perturb transform after pre-training on $50\%$ of CIFAR-10. We used $\sigma = 10^{-4}$ and repeated this experiment with different values of $\lambda$. We plotted the test accuracy during training on all of the data in Figure 5. It can be seen that the method is effective and leads to even better performance than the randomly initialized model.

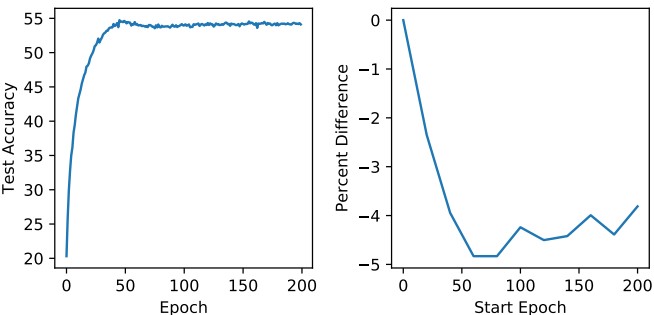

Figure 3: **Left**: Test accuracy while training on half of CIFAR-10. **Right**: Plot of test accuracy damage, as percentage difference from random initialization, against number of warm-starting epochs.

Figure 4: Training time vs Test accuracy for randomly initialized and warm-started models

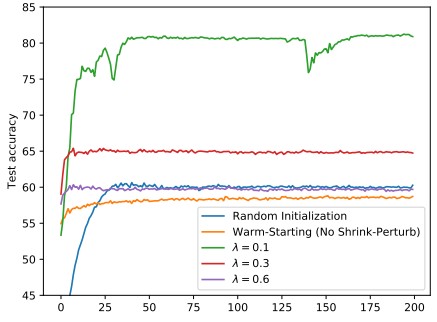

Figure 5: Test accuracy during training a ResNet-18 with SGD with warm starting and Shrink-Perturb and without it

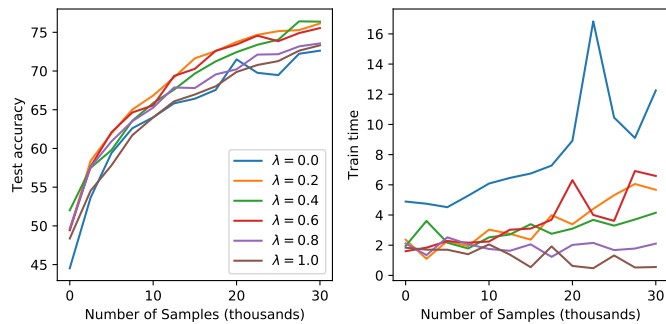

Figure 6: Test accuracy during training a ResNet-18 in the online setting while applying Shrink-Perturb method with different values of $\lambda$.

In the online setting, we applied the Shrink-Perturb transform every time a new batch of data is added. To reduce computation cost, we add data in batches of 2500 samples. The result for $\sigma = 10^{-4}$ and different values of $\lambda$ is plotted in Figure 6. Figure 6 matches Figure 7 of the original paper. The result of applying the Shrink-Perturb method in the offline setting is not reported in the original paper.

To assess the impact of shrinking weights on the model's performance, we fit different models to CIFAR-10. Then, we shrink the weight with different values of $\lambda$ and evaluate the accuracy. Similar to the paper, we train ResNet18 and an MLP with ReLU activation with and without bias. In addition, we also train an MLP with Tanh activation with and without bias. The result is shown in Figure 7, which corresponds with Figure 6 of the paper. The only difference in our findings with the original paper's is that we observe classifier performance damage for MLP with ReLU for $\lambda > 0.6$. Though for $\lambda > 0.8$ the damage is negligible. Also, note that shrinking the weights of an MLP without bias and ReLU activation only scales the final output, which does not affect the output labels. Therefore its immunity to shrinkage is expected. The more interesting result is that even for ResNet-18 or MLP with Tanh activation, the test accuracy is not significantly damaged for $\lambda$ values greater than 0.2.

In order to explain why the Shrink-Perturb method is effective, the original paper compares the average gradients over the first and sec-

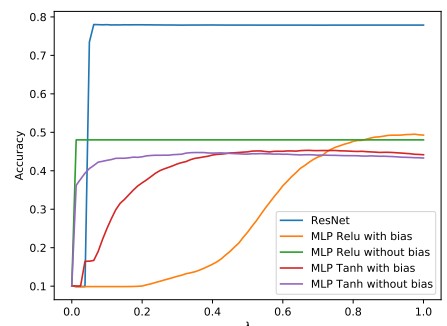

Figure 7: Test accuracy of trained models for different shrinkage coefficients $\lambda$.

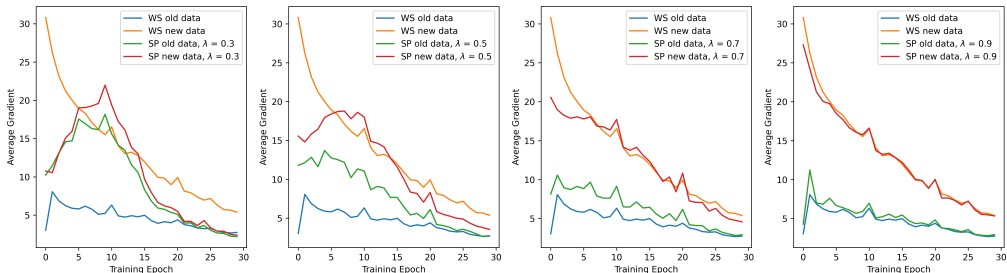

Figure 8: Average over old data or new data of L2 norm of gradient with respect to model parameters. WS refers to warm starting without and SP to warm starting with Shrink-Perturb.

ond half of the dataset during the training of the warm-started model in the offline setting. In particular, a ResNet-18 is first trained on half of CIFAR-10. Warm-starting from the pre-trained model, the model is trained on the full dataset while measuring the average gradient over the first and second half of the dataset simultaneously. It is observed that the gradient for the first half, which the model was pre-trained on, is substantially lower than for the second part. However, applying the Shrink-Perturb transformation eliminates this difference. We reproduced this experiment with some slight modifications. In particular, instead of averaging the gradient over part of the dataset after each batch, we did it at the beginning of each epoch. We plotted these values in Figure 8. Our result matches Figure 5 of the original paper. In particular, we confirm that the Shrink-Perturb method successfully eliminates the gap between the gradients.

## 2.4 Effect of Data Augmentation

Data augmentation is widely used in practice. However, it is not used for the experiments in the original paper. Therefore, we decided to assess its impact on the generalization gap for warm starting training.

We perform our experiments on ResNet-18 and CIFAR-10. To augment the data, we first pad the image with 4 pixels on each side and then randomly crop it back to 32x32. We then perform a random horizontal flip with probability $0.5$. We also apply color jitter with brightness, contrast, and saturation factor equal to $0.25$. Finally, we also apply small random rotations. All experiments were done with SGD and a learning rate equal to 0.001 in order to make the setup consistent with previous warm start experiments. The results are reported on Figure 9 .

It can be seen that applying the augmentation mitigates the warm-starting gap. We allow the models to train for 350 epochs. However, because the learning rate is low, the models are not fully converged even after 350 epochs. We did not continue the training because of resource limitations. However, it is visible that warm-starting with data augmentation can achieve good performance faster than training from scratch.

To explain the effectiveness of the Shrink-Perturb solution, the original paper's authors looked at the differences of the gradient norm for the first and the second part of the dataset, which is heavily reduced after applying Shrink-Perturb (as shown in Figure 8). Following the same direction, we checked if applying data augmentation can affect the difference as well. It can be seen in Figure 10 that, similar to the shrink perturb method, the gradient norm difference is also mitigated when using data augmentation.

It is clear that applying data augmentation prevented overfitting. The original warm start setup has a large divergence between train and test accuracy from the beginning of the training. On the contrary, the model trained with data augmentation has close performance on train and test datasets. We leave the investigation of other overfitting prevention techniques' effects as future work.

Additionally, we note that data augmentation usually slows down the convergence and it cannot be applied to every task since for some types of data transform set cannot be defined. Due to the limits of this report, we also leave the careful comparison between data augmentations and Shrink & Perturb as future research in this area.

## 2.5 Warm-Starting Gap in Transfer Learning

Deep learning models require large training sets to perform well. This presents a problem in many practical cases where only limited data is available, and acquiring additional data is expensive. This has encouraged the use of transfer learning [7; 8; 9]; the practice of warm-starting from a model trained on a different dataset.

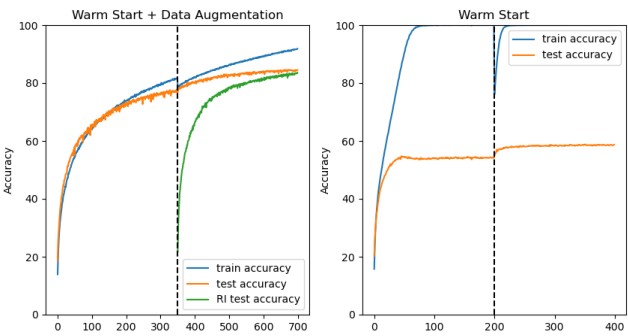
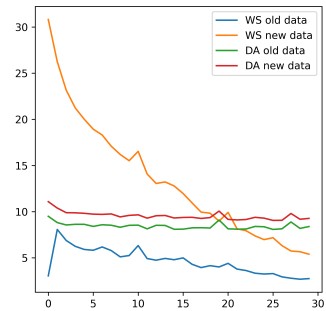

Figure 9: Test and train accuracy of models in warm start setting with and without data augmentation. RI refers to Random Initialization.

Figure 10: Gradients with respect to the first and the second half of the dataset, DA refers to Data Augmentation

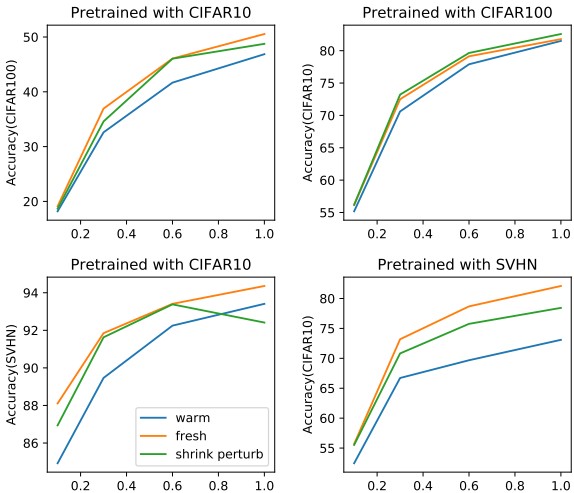

Figure 11: Test accuracy against the available portion of data $p$ for training a model on one dataset with warm-starting from another dataset.

To investigate whether a similar gap is observed in transfer learning, we trained a ResNet-18 model on one dataset and used the pre-trained weights to warm-start training on a different dataset. We performed this experiment for all pairs of CIFAR-100, CIFAR-10, and SVHN datasets. To also investigate the effect of the amount of the data available, we also considered subsets of these datasets where only a fraction $p$ of data is available. More accurately, for every two datasets and $p \in \{0.1, 0.3, 0.6, 1.0\}$, we chose a random subset from each dataset containing $p \times n$ data points, where $n$ is the total number of data points in that dataset. We performed the described transfer learning experiment for these subsets and recorded the final test accuracy. To assess the effect of warm-starting and the Shrink-Perturb method, we plotted the final test accuracy of each of the described three settings (random initialization, warm starting, and warm-starting with Shrink-Perturb) with respect to $p$ in Figure 11. In this experiment, we used Adam as our optimizer. This figure corresponds with Figure 9 of the original paper.

At each part of the training, we train our models for 200 epochs. When training CIFAR-100 starting from weights of a model trained on SVHN or CIFAR-10, the last layer is initialized randomly because of the mismatch in the number of classes. It can be seen that, as mentioned in the paper, the warm-starting gap exists in the transfer learning settings as well, and that it is worsened when the amount of data available is increased. Furthermore, the Shrink-Perturb method proves useful in this setting, as well.

## 3  Methodology

In this section, we define the setting we used for our experiments. There was no available code for the original paper, and we implemented everything from scratch. We use the PyTorch framework for the implementations.

### 3.1 Model descriptions

Most of the experiments are performed using ResNet-18 [1]. Some experiments are also performed on a Multi-Layered Perceptron (MLP) and Logistic Regression. We detailed the structure of each of these models below.

- **ResNet-18:** We used an implementation of ResNet-18 tuned for CIFAR-10 dataset. We used the code from `https://github.com/huyvnphan/PyTorch_CIFAR-10`. In all experiments, batch normalization [10] was enabled.
- **MLP:** The MLP has three hidden layers, each of which has 100 neurons. Either ReLU or Tanh was used as the activation function. Unless explicitly stated, the bias term is added.
- **Logistic Regression:** We implement Logistic Regression as a Multi-Layered Perceptron with no hidden layers.

We used either Adam or SGD optimizers for training the models. More accurately, we use Adam in Table 1, Figure 2, Figure 6, and Figure 11. In all other experiments, we use SGD. We use $0.001$ learning rate and batches of size $128$. Unless otherwise stated, we used SGD without momentum and without weight decay. In cases where momentum was used (such as in Table 1), the value of momentum was set to $0.9$. The Adam optimizer was used with default parameters from PyTorch's implementation, namely $\beta_1 = 0.9$, and $\beta_2 = 0.999$.

### 3.2 Datasets

Same as the original paper, we perform experiments on CIFAR-10, CIFAR-100, and SVHN datasets. We normalize each of the RGB channels by the mean and standard deviation of that channel in the CIFAR-10 dataset. Except for the data augmentation experiments, we do not apply any data augmentation.

### 3.3 Hyperparameters

We used hyperparameters stated in the original paper in most of our experiments. In cases where we deviated from the reported values, mostly due to computational resource and time limitation, we have reported them in the text where we described the experiment. In case a hyperparameter is not reported in the original paper, we either communicated with the authors to ask the hyperparameters, pick a value making reasonable assumptions, or try out different values and report the result for all of them. In all these cases, we clarified the parameter we used in the text.

### 3.4 Experimental setup

We ran our experiments on both public cloud infrastructure, such as Google Colab and private GPUs that were available to us. Therefore the infrastructure varies between different experiments. Our implementations for all the experiments in this work is available in the Supplementary Material and also in `https://github.com/CS-433/cs-433-project-2-fesenjoon`.

## 4 Communication with Authors

In the original paper [11], it was not clear what convergence condition was used to stop the training. Therefore, We communicated with the authors via email and asked them to explain the convergence condition used in the experiments more clearly. They stated that convergence happens when the training accuracy reaches 99%. However, for reproducing the Table 1 which also uses simpler models like Logistic Regression and MLP, it is not possible to reach 99% accuracy. They clarified that in this scenario, the convergence condition is met when the training accuracy stops improving.

## 5 Conclusion

We have verified the existence of the generalization gap in certain training settings. Additionally, we have confirmed that the introduced Shrink-Perturb method can be effective in removing this gap. We did this by repeating experiments of the original paper and performing some experiments of our own. However, we also encountered cases where we were not able to reproduce the warm-starting gap or where the Shrink-Perturb method was not very successful. In addition, we reproduced several experiments to investigate the effect of hyper-parameters, such as learning rate, on this phenomenon. Finally, we have shown that applying data augmentation can also help to remove this gap. To allow others to reproduce our results, we have detailed our experiments and have released our code.

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
