# OpenReview forum: "Reproducibility Report for "On Warm-Starting Neural Network Training""
_ML_Reproducibility_Challenge/2020 — RC2020_

### Official Review · AnonReviewer3 · 2021-02-17
**Well structured report, but a lot of inconsistencies in experiments. Not clear if reproduction wasn't proper or if the original paper results are bogus**

**Rating:** 5
**Confidence:** 4

**Review:**

This report is reproducing most of the experiments from the paper "On Warm Starting Neural Network Training". While the general trends are visible in the reproduced results, there are many details that are not the same. As the reproduced paper doesn't provide confidence intervals in their figures (which the original paper does) and they show test instead of validation results, it is not very easy to compare results. Equivalent experiments in most of the examples achieve different max accuracy. Even more concerning is that gap that is shown in the very first experiment in order to motivate the work, is almost not existing in Table 1 of reproduction. Other experiments also show unexpectedly good performance of warm start models in the reproduced results. Overall, experiments results within the report show inconsistent behavior and instability.

The report is well written and organised and it contains a summary section at the beginning. Report authors clarified questions with original authors. The link to the implementation doesn't work. I've searched the repository and that project is deleted.

I like the report, but I am giving grade 5 because the results are not matching the original paper results, and since they are less extensive and elaborative and with no confidence intervals, I have more confidence in the results of the original paper. Additionally, the link to the code doesn't work, so it's not possible to see the setup of the reproduced experiments.

Below are comments related to different experiments in the order in which they appear in the report:
- Experiment in Figure 1: Report authors use only 200/400 epochs instead of 350/700 epochs used in the original paper, but they are able to show the same effect. It is good that even with fewer time resources, we can show the same effect.
- Table 1: While the original paper gives us validation accuracies, the report authors give us training and test accuracies, so it's not comparable. For LR, there is no gap according to Table 1, but similar is true for original Table 1. However, while in the original Table 1 gap is obvious between random init and warm start, in the reproduced study, that's not true. Most of the gaps are within one standard deviation and gaps are inconsistent, meaning that in some cases warm start is better. Also, test results here are much worse than original validation results. Did they train for the same number of epochs? The difference is too big to be explained by validation vs test set difference.
- Figure 2: there is no confidentiality interval that exists in the original paper. As authors of the report saw that SVHN behaves differently, they added additional experiments here, which show slightly different behavior and require a convergence threshold of 99.9%. I would suggest changing the scale of Figures 2 b) and c) to show only information above 80% or 85 % of accuracy so that it is easier to see the gap.
- Results in Figure 4 are different than in the original study. In this report, warm start models are performing consistently better than in the original paper, so we could see there was almost no gap visible in Table 1, and in Figure 4 performance of warm start is better than the random start, even though that's not true in the original paper. Again, the report deals with test errors while original papers worked with validation errors, but that shouldn't affect the conclusion. It is also interesting to note that the report is able to achieve better results on the test dataset with a few warm start experiments than the original paper achieves on the validation dataset with any model. That is suspicious to me.
- While the original paper has confidence intervals for Figure 4, that is not given in reproduction (Figure 3) and the patterns show small but visible differences, which might be explained with confidence intervals. Again, the test accuracy that is achieved by the report is up to 5% higher than in the original paper.
- What is the x-axis of Figure 5? Is it the number of epochs?
- Figure 6 (matches original Figure 7): Behavior is inconsistent for different lambda values. No pattern can be spotted in this figure, except that for lambda=0 train time is strongly the highest. It is very strange that for lambda=0 training time is twice bigger than expected for around 25 thousand examples.
- In general, results reported in those figures are less smooth, which is probably because the information is displayed in more coarse grain, but it may also be due to the instability of the models or issues with reproducibility. However, it is not clear from the report which one it is, and it seems it's rather from the former.
- Authors add experiments on warm start with augmented data in which warm start slightly outperforms random initialization. However, in some of the repeated experiments, we can see similar behavior in the report, so it looks like warm start performs better throughout this report and not because of the data augmentation.
- Figure 11: Why does shrink-perturb go down after on the graph in the bottom left corner when x > 0.6? Most of the results in this figure are inconsistent with the original paper as they show that fresh has the best performance, but results are inconsistent. Also, on the contrary from the rest of the report, here we can see that warm has the worst performance in all the cases.

**Familiar With The Original Paper:**

I have read the original paper

**Reproducibility Summary:**

Report has summary

---

### Official Review · AnonReviewer1 · 2021-02-26
**a relevant review toward the original paper with minor mistakes according to report tempelate**

**Rating:** 9
**Confidence:** 5

**Review:**

In the proposed paper, one of the main pros is that you've plotted and calculated each relation that was measured in the original paper and obtained mainly near or the exact values toward the original paper. But some minor issues showed off during reviewing your paper.
First of all, you've forgotten to blind your names and affiliations, not a major problem but it might affect reviewers' ideas sometimes. After that, you've also forgotten to numerize lines, so I have to mention them using paragraphs' titles.
According to the "What was easy" paragraph, you'd better re-write "since many of the parameters are reported in the original paper" into "since many of the parameters ***were*** reported in the original paper." There also was so good that you've explained all the conditions you've put your dataset under test, like the optimizers, loss functions, etc. and I guess you've tested as far as I know enough conditions to explain your model and results.
As it seems, the original paper didn't do the ***data augmentation***, but it's so good to multiply the amount of data to exploit enough accuracy.
In the "effect of hyperparameter" paragraph, you've made a typo and wrote ***vales*** instead of ***values***, in the "We iterate over all pairs for these vales" sentence.
In the "effect of data augmentation" paragraph, you've mentioned that "However, because the learning rate is low, the models are not fully converged even after 350 epochs", although you've previously mentioned that your model converged to 99% of accuracy, so it wasn't crystally clear for me. In the sentence "the original paper’s authors look at the difference", it would better to re-write ***look*** to ***looked***, and ***difference*** to ***differences***. In the last sentence of this paragraph "Due to the limits of this report, we also leave the careful comparison between data augmentations and Shrink & Perturb as future research in this area", it would be better to explain ***limits*** more.
The link you've attached as "https://github.com/CS-433/cs-433-project-2-fesenjoon." didn't work properly, I couldn't find your project, according to the link you've attached in the ***Experimental setup*** paragraph.
But in total, you've represented your method very well.

**Familiar With The Original Paper:**

I have read the original paper

**Reproducibility Summary:**

Report has summary

---

### Official Review · AnonReviewer2 · 2021-03-01
**Review: Strong Accept**

**Rating:** 9
**Confidence:** 5

**Review:**

1. In this report the authors have tackled the following reproducibility claims from the original paper:
  - Warm-starting a Neural Network training has poorer generalization compared to training with new data + old data from scratch
  - Application of Shrink+Perturb technique proposed by authors to rectify and improve the warm-starting generalization.
2. The authors have implemented the code by themselves and tried various hyperparameters as mentioned by the original authors. They communicated with the original authors to clarify some of the implementation details mentioned in the paper like when to stop for training convergence, hyperparameters to tune, etc. However, the public link for the code implemented by the report's authors is not accessible.
3. In addition to implementing the code, the authors have also tried other techniques that might help with the warm-starting problem such as data-augmentation, early-stopping, regularization etc.
4. In the results and discussions part of the report, the authors have described the implementation details of the project. The authors have reported that warm-starting definitely has a generalization gap, shrink and perturb method is effective in reducing the gap.


**Familiar With The Original Paper:**

I have read the original paper

**Reproducibility Summary:**

Report has summary

---

### Decision · Program_Chairs · 2021-03-31

**Decision:**

Accept

**Comment:**

Selected for ReScience-C Journal Publication.

The paper is a strong reproduction effort as they perform analysis on more datasets, augmentations, etc. The report goes well and beyond, which is highly commendable.